# OpenReview forum: "Falcon: A Cross-Modal Evaluation Dataset for Comprehensive Safety Perception"
_ICLR.cc/2026/Conference — ICLR 2026 Conference Withdrawn Submission_

### Official Review · Reviewer_6Hmy · 2025-10-23

**Soundness:** 2
**Presentation:** 2
**Contribution:** 2
**Rating:** 4
**Confidence:** 4

**Summary:**

This paper shows that visual context is crucial for moderating visual question answering, yet is often overlooked. To address this, this paper introduces Falcon, a 57,515-pair vision–language safety dataset spanning 13 harm categories, with explicit labels for harmful attributes in images, instructions, and responses, as well as category tags and rationales. Moreover, this paper also presents FalconEye, a Qwen2.5-VL-7B evaluator fine-tuned on Falcon. FalconEye reliably detects harmful content on three benchmarks.

**Strengths:**

1. The writing is good and easy to understand.
2. Based on the experiments, the detector trained on the authors’ dataset outperforms existing baselines, even beating GPT-4o.
3. The experiments are comprehensive, covering multiple baselines and evaluation datasets.

**Weaknesses:**

1. In Section 3.1 (Generating VQA pairs), the authors say “we manually carry out several rounds of cleaning on the obtained VQA pairs to filter out low-quality and disorganized data. In total, we obtain 57,515 VQA pairs,” but they omit key details: How many annotators were there? What are their backgrounds? What guidelines were used to filter low-quality samples? The same issue appears in the “Human Safety Annotations” stage.
2. In the provided dataset, the training set is fully auto-labeled by Qwen2.5-VL-72B-Instruct-AWQ, with no necessary quality control.
3. The contribution is fairly incremental: it aggregates several existing datasets, uses an open-source model for auto-labeling, then manually labels a 1,800-sample test set, and finally fine-tunes models on this dataset to compare with existing baselines.

**Questions:**

See Weaknesses.

---

### Official Review · Reviewer_JGfq · 2025-10-27

**Soundness:** 2
**Presentation:** 3
**Contribution:** 1
**Rating:** 2
**Confidence:** 4

**Summary:**

The submission
- constructs a training set for safety classification of VQA inputs and outputs by pooling input samples from existing datasets, generating responses and auto annotating inputs and responses from existing open source VLMs.
- Additionally it provides a test set of human annotations for 1.8 k samples.
- It fine-tunes Qwen-2.5-VL-7B on the constructed data and shows improvements in safety classification on the provided test set, as well as Beavertails-V and VLGuard datasets.

**Strengths:**

- The paper reasonably demonstrates that distilling the annotations from the large Qwen-2.5-VL-72B-AWQ on the pooled training set improves the performance of Qwen2.5-VL-7B in safety classification for input and responses on the given VQA datasets.
- It provides a set of 1.8k samples with human annotations

**Weaknesses:**

- The proposed dataset has little novelty, it is pooling samples from existing datasets and generating responses and labels from existing open source models.
- The labelling scheme of the dataset is lacking depth and nuance. E.g.
    - The dataset completely ignores compositional effects between text and images in the input label, where the label of the VQA pair is different to the label of the individual modalities (see. [1-4] for recent examples discussing this)
    - The dataset does not seem to have a notion of inputs that are related to harmful topics but are not necessarily seeking a harmful response (see [4, 5] for recent discussion).
- At least parts of the harm categories are poorly motivated, e.g. the decision “both "Child Abuse" and "Animal Abuse" pertain to forms of physical or psychological harm. We merge them under the broader category of "Abuse", which also encompasses other potential types of abusive behavior.”  is concerning, as both of these harm categories have dramatically different levels of risk in real world deployments and will almost certainly require different policy treatment.
- In summary, I don’t think the dataset makes a meaningful conceptual or practical contribution to the community.
- No analysis of the quality of the human annotations. E.g. what was the level of agreement between multiple human graders and how well are the auto-annotations from  Qwen-2.5-VL-72B-AWQ aligned with the human annotations on the test set?

[1] Wang, Siyin, et al. "Safe Inputs but Unsafe Output: Benchmarking Cross-modality Safety Alignment of Large Vision-Language Model." arXiv preprint arXiv:2406.15279 (2024).

[2] Röttger, Paul, et al. "MSTS: A Multimodal Safety Test Suite for Vision-Language Models." arXiv preprint arXiv:2501.10057 (2025).

 [3] Lee, Youngwan, et al. "HoliSafe: Holistic Safety Benchmarking and Modeling with Safety Meta Token for Vision-Language Model." arXiv preprint arXiv:2506.04704 (2025).

 [4] Palaskar, Shruti, et al. "VLSU: Mapping the Limits of Joint Multimodal Understanding for AI Safety." arXiv preprint arXiv:2510.18214 (2025).

 [5] Yuan, Yuan, et al. "From hard refusals to safe-completions: Toward output-centric safety training." arXiv preprint arXiv:2508.09224 (2025).

**Questions:**

- Human annotations
    - What is the agreement between auto annotations and human annotations -> Could be measured on the test set.
    - What is the agreement across humans annotators?
- Baselines
    - How were GPT4o and Qwen2.5-VL-7B prompted?
    - How well would they perform with simple few-shot examples from the training set or simple prompt-optimization techniques (e.g. TextGrad or GEPA)?

---

### Official Review · Reviewer_KDco · 2025-10-28

**Soundness:** 2
**Presentation:** 2
**Contribution:** 2
**Rating:** 2
**Confidence:** 4

**Summary:**

This paper presents Falcon, a large-scale cross-modal evaluation dataset for safety perception in multimodal large language models (MLLMs). Falcon contains 57,515 visual question answering (VQA) pairs annotated across 13 harm categories, with explicit labels on images, instructions, and responses. The authors also introduce FalconEye, an evaluator model fine-tuned on this dataset, and demonstrate its effectiveness on the Falcon-test, VLGuard, and Beavertail-V datasets. FalconEye is shown to outperform several established baselines, including open and closed-source models, in accuracy across multiple harm dimensions. The dataset, model, and evaluation methodology are released with a focus on enhancing safety assessment rigor for MLLMs.

**Strengths:**

- Comprehensive Cross-Modal Dataset Design: Falcon represents a good contribution as a large, fine-grained, multimodal dataset targeting safety evaluation in VQA scenarios. Unlike some predecessors, it provides per-instance annotations for harmfulness across images, prompts, and model responses, each with explicit harm categories and justifications.
- Coverage and Scope: Falcon covers 13 differentiated harm types, consciously merging overlapping categories from prior works to provide clearer, operationally feasible definitions.
- Rigorous Annotation Pipeline: The dataset employs a multi-stage semi-automated/human- in-the-loop curation, with manual annotation for the Falcon-test subset.

**Weaknesses:**

- Limited Novelty / Incremental Contribution: Amid many recent multimodal safety benchmarks, Falcon appears incremental. Falcon contributes no original data. The samples are aggregated/reprocessed from existing open-source datasets (e.g., SPA-VL, JailBreakV-28k, HADES). There is no new scenario, data collection protocol, or annotation methodology introduced.
- Class Imbalance: Several harm categories, such as "Unlicensed Advice" and "Adult Content," are underrepresented in the dataset. While FalconEye shows reasonable performance even on rare classes, Table 7 and associated text indicate that accuracy drops significantly for minority classes. The paper acknowledges these challenges, but no mitigation methods are proposed.
- Reliable Evaluation Metric: The paper leans almost entirely on accuracy, which could be misleading for imbalanced and multi-label tasks. It does not report precision, recall, F1, or results across multiple random seeds.
- Annotation Process Lacks Detail: While Section 3.1 claims consensus and guidelines for annotators, no quantitative metric is reported to establish reliability.

**Questions:**

1. Can the authors provide quantitative measures for the manual annotation process of the Falcon dataset?
2. What specific steps, if any, were taken to mitigate the underrepresentation of rare harm categories in model training?

---

### Official Review · Reviewer_RFZL · 2025-11-01

**Soundness:** 3
**Presentation:** 3
**Contribution:** 3
**Rating:** 4
**Confidence:** 4

**Summary:**

In this paper, the authors proposed Falcon, a large scale vision-language safety dataset. A fine-tuned model FalconEye was also introduced by tuning Qwen model on the Falcon dataset. Multiple experiments show that FalconEye demonstrated good accuracy on Falcon-test and two other benchmarks.

**Strengths:**

1. The current draft is addressing a very critical issue of safety evaluation for Multimodal Large Language Models (MLLMs) where visual information is crucial but usually overlooked in the existing safety evaluation of Large Language Models (LMMs).
2. The proposed data curation and model training/fine-tuning process also make sense and are both technically sound to me. The annotation of all three parts of the VQA triplet allows for a nuanced understanding of safety that is missing from benchmarks that only label responses or inputs.
3. Extensive experiments were conducted to show that FalconEye is a effective model for evaluating the safety of MLLM-generated content.
4. Writing is good and easy to follow.

**Weaknesses:**

1. My main concern is that the current data curation process heavily rely on MLLMs themselves. While it automate the data curation process and make it more scalable, it is a bottleneck and capped by the capability of the models used.
2. Even though FalconEye showed good performance on binary harm detection in Table 3, it seems very weak on more fine-grained classification tasks based on the results from Table 7.
3. Since FalconEye was trained on Falcon dataset and Falcon-test is more of a in domain data, I am not sure the comparison to other models are fair enough. Will FalconEye work well on out of domain data as well?

**Questions:**

Please refer to the paper weakness section for more details and provide more justifications accordingly.

---

### Note · Authors · 2025-11-26

I have read and agree with the venue's withdrawal policy on behalf of myself and my co-authors.